# Kidney and kidney tumor segmentation using a two-stage cascade framework

Chaonan Lin[1], Rongda Fu[2], and Shaohua Zheng[1,*]

[1] College of Physics and Information Engineering  Fuzhou University
[2] School of Mechanical engineering and Automation  Fuzhou University
sunphen@fzu.edu.cn

**Abstract.** Automatic segmentation of kidney tumors and lesions in medical images is an essential measure for clinical treatment and diagnosis. In this work, we proposed a two-stage cascade network to segment three hierarchical regions: kidney, kidney tumor and cyst from CT scans. The cascade is designed to decompose the four-class segmentation problem into two segmentation subtasks. The kidney is obtained in the first stage using a modified 3D U-Net called Kidney-Net. In the second stage, we designed a fine segmentation model, which named Masses-Net to segment kidney tumor and cyst based on the kidney which obtained in the first stage. A multi-dimension feature (MDF) module is utilized to learn more spatial and contextual information. The convolutional block attention module (CBAM) also introduced to focus on the important feature. Moreover, we adopted a deep supervision mechanism for regularizing segmentation accuracy and feature learning in the decoding part. Experiments with KiTS2021 validation set show that our proposed method achieve Sørensen-Dice scores of 0.9304, 0.5729 and 0.563 for kidney, masses (tumor and cyst) and kidney tumor, respectively.

**Keywords:** cascade framework · kidney/tumor segmentation · deep learning

## 1 Introduction

Kidney cancer is one of the most aggressive cancer, which have 40 0000 growth numbers and high fatality rate of 40%. Renal cell carcinoma (shorted as kidney tumor) and renal cysts (cysts for short) are the most common diseases that cause to it. The cysts are formed in the kidneys with age, and won't easily bring about injury while tumors often pose high risks to human health.

Studies have shown that tumors are more susceptible to effective treatment if they are detected at earlier stage. However, these tumors may grow into a large size before being detected[1]. Therefore, early accurate diagnosis can effectively improve the survival rate of kidney cancer patients. The success of such studies relies on the computed tomography (CT) technology, which can provide high-resolution images with good anatomical details. Combined with the great potential information from medical images, such as the location, shape and size of

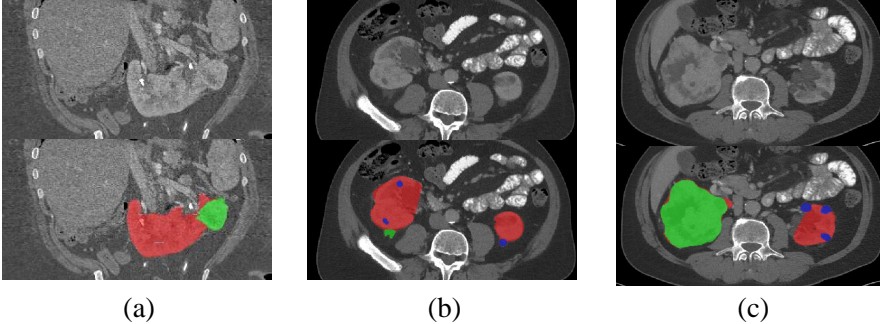

(a)                              (b)                              (c)

**Fig. 1.** Illustration of sample segmented images from three patients in the KiTS21 dataset. The first row is the transverse plane, and the second row is with the labels. Red, green and blue denote kidney, kidney tumor and cyst, respectively.

the kidney and tumor, radiologists can know about disease severity and progression, then make a more accurate clinical decision. Hence, accurate segmentation of kidney and kidney tumor is an essential step for radiomic analysis as well as developing advanced surgical planning techniques. The segmentation of kidney, kidney tumor and cyst are usually manually marked by radiologists. However, manual segmentation is a time-consuming and tedious task due to the hundreds slices of CT. Moreover, the label results strongly depend on the experience of radiologists and prone to errors. In order to reduce the burden of manual works and improve segmentation accuracy, automatic segmentation of kidney, kidney tumor and cyst has become a new demand.

Deep learning (DL) technology has been widely applied in the medical image field and play an essential role in kidney and kidney tumor segmentation works. Methods based on deep learning are categorized into two: one-stage and two-stage method. One-stage methods [2–7] are designed to predict the multi-class results directly from whole images. Guo et al.[5] proposed an end-to-end model based on residual and attention module. Residual connection was added to each convolutional layer to generate clearer semantic features. Skip connection was also used in attention module to make the decoder focus on the segmentation target. Zhao et al.[6] presented a multi-scale supervised 3D U-Net to segment kidneys and kidney tumors from CT images. Multi-scale supervision was adopt to obtain more accurate predictions from deep layers. Sabarinathan et al.[7] presented a novel kidney tumor segmentation method. This work introduced supervision layers into the decoder part, and coordinate convolutional layer was utilized to improvise the generalization capacity of the model.

Two-stage methods [8–13] aim to solve the imbalance problem between foreground and back-ground. Those methods firstly detect the volume of interest (VOIs), then segment the target organs from the VOIs. A typical two-stage method was proposed by Cheng et al.[8], they employed a double cascaded framework, which decomposed the complex task of multi-class segmentation into two

simpler binary segmentation tasks. In the first step, the region of interest (ROI) including kidney and kidney tumor is extracted, and then segment the kidney tumor in second step. However, these works still suffer from several anatomical challenges. First, the low contrast between kidney and nearby organs, the unclear boundaries and heterogeneity of tumor, all make accurate segmentation become more difficult, as shown in Fig .1 (a) and Fig .1 (b). Second, Fig 1. (c) indicates that kidney tumors and cysts exhibit various size, shape, location and number from different patients.

To address the above challenges and improve the segmentation performance of unbalanced kidney and tumor datasets, we proposed a two-stage framework to obtain kidney and masses (kidney, cyst), respectively. In this framework, the complex multi-class segmentation task is transformed into two simplified subtasks: (i) locating the kidney region and segmenting the kidney, (ii)segmenting the kidney tumor and cyst in the kidney. The Kidney-Net applied in the first stage is modified based on a normal 3D U-Net, while our core works mainly focus on masses segmentation in the second stage. In the second stage, a fine segmentation network Masses-Net is trained based on the cropped kidney region obtained in the first stage to segment tumors and cysts. In order to leverage more useful features, a multi-dimension feature (MDF) module is utilized to learn more space and context information. Meanwhile, convolutional block attention module (CBAM) also applied to focus on the important feature. Finally, a deep supervision mechanism was also used in the decoding, which works as a regularizing role in segmentation accuracy and feature learning.

## 2   Methods

In this section, we mainly introduce our method for kidney and kidney masses (tumor and cyst) segmentation. The proposed two-stage segmentation framework is illustrated in Fig 2. The framework consists of two phases: the first stage for kidney segmentation and the second stage for masses (tumor and cyst) segmentation.

In the first stage, the pre-processed CT images are fed into a kidney segmentation network, which named Kidney-Net. The output of Kindey-Net is a coarse segmentation result of overall kidney and is binarized to produce an overall kidney mask. The mask is applied for boundary coordinates and crop volume of interest (VOI). The cropped VOI is the input of the second stage, a Masses-Net is used for tumor segmentation and cyst segmentation.

In the training processing, the two networks are trained individually due to the different input patches. In the testing processing, two individual results of Kindey-Net and Masses-Net are fused via a union method, which add the two prediction results directly. Then the merge result is refined by a post-processing method.

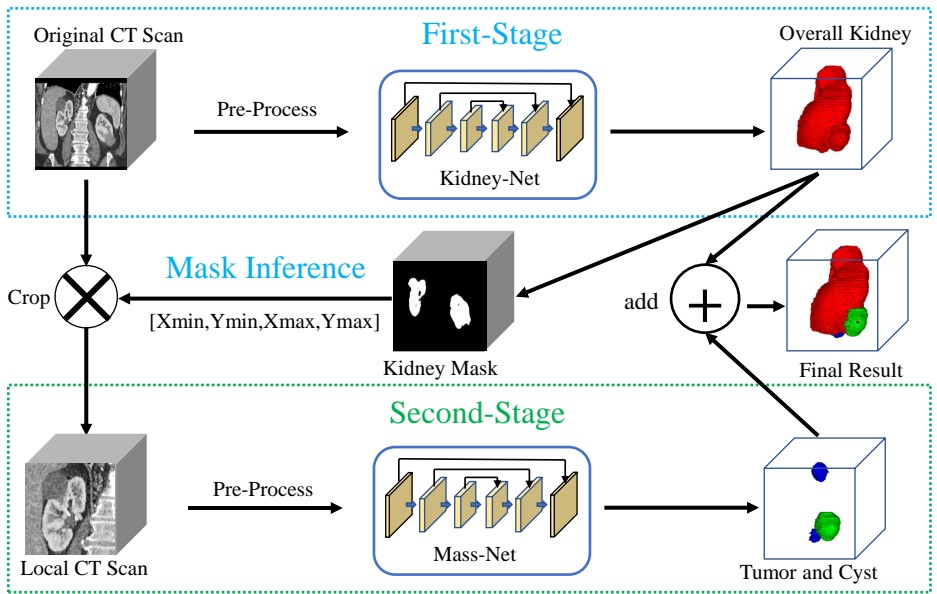

**Fig. 2.** Schematic of the workflow of the proposed two-stage segmentation framework.

### 2.1   Kidney-Net

In our method, we concatenated two networks to segment the kidney, kidney masses (tumor and cyst) respectively. The Kidney-Net in the first stage is used for kidney segmentation. As shown in Fig. 3(a), Kidney-Net is a u-shaped network which only contains three pooling layers. And the encoder/decoder blocks are all regular convolution block, which is composed of two convolution layers with batch normalization (BN) and rectified linear unit (ReLU). Due to the Kidney-Net is trained to predict the probabilities of every voxels belong to kidney, Dice loss function is utilized for the voxel level classification task and be formulated as:

$$\mathcal{L}_{kidney} = 1 - \frac{2 * \sum_{i=1}^{2} (r_i * t_i)}{\sum_{i=1}^{2} (r_i + t_i) + \theta} \tag{1}$$

Where $r_i$, $t_i$,$i \in 0,1$ is the segmentation kidney result and target kidney mask, respectively, and $\theta$ is a smooth term to avoid division by zero.

### 2.2   Masses-Net

The architecture of Masses-Net is shown in Fig.3(b). Masses-Net is similar to the Kidney-Net. For the purpose of utilizing more global features, the encoder of Masses-Net contains four pooling operations and four encoder blocks. The input

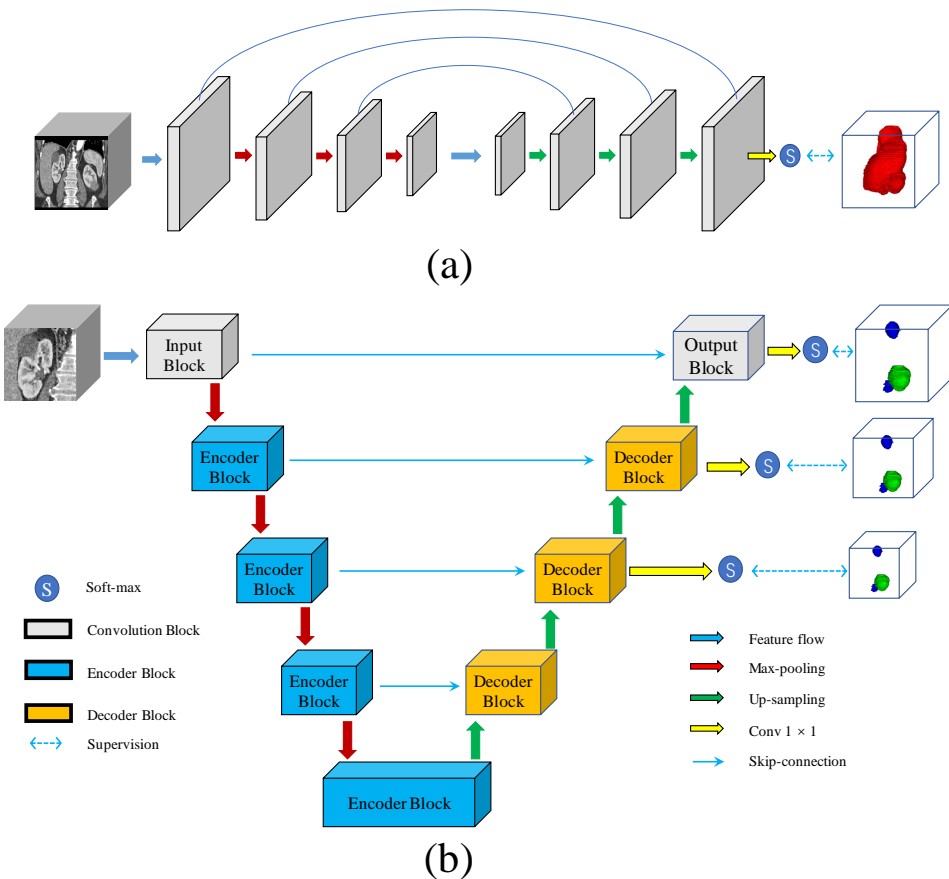

**Fig. 3.** Overview of the proposed segmentation network. (a) The basic architecture of the Kidney-Net (b)The basic architecture of the Masses-Net

and output blocks are regular convolution block, which used for generate low-level feature maps. Then, the feature maps generate by input block are fed into four successive encoder blocks to obtain global features. The decoder is used for target segmentation and working in a coarse-to-fine pattern, which also contains four decoders. Finally, the output of the last three decoder blocks is fed into Soft-max activation function for the tumor and cyst prediction. Moreover, deep supervision scheme is applied. The key components are illustrated as follow.

**Residual Connection mechanism** Considering the problems related to over-fitting and vanishing gradient, residual connection is incorporated to maintain more spatial and contextual information and make the learnable network parameters increasingly effective [14]. Fig. 4 (a) shows the input/output block, which

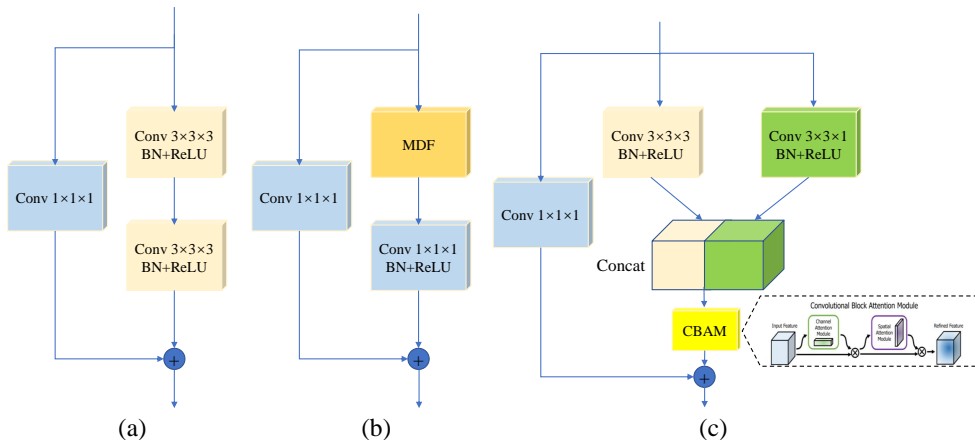

**Fig. 4.** (a) The input/output block, (b) the encoder/decoder of proposed network, (c) multi-dimension feature (MDF) module.

is similar to a regular residual block. Combining the residual connection, the network can achieve a better result. Fig. 4 (b) shows a encoder/decoder block of our network. In this block, a multi-dimension feature (MDF) module is applied to replace the regular convolution layers. And the architecture of MDF is shown in the Fig. 4 (c), residual connection also adopted in the module.

**Multi-dimension Feature Module** Due to the 3D kernel can be used to capture more spatial information, 3D neural network is adopted for segmentation tasks in medical image. However, some important characteristics frequently appear in the x/y plane and be ignored by the 3D kernels. Inspired by the anisotropic convolutional [15], multi-dimensions feature (MDF) module is proposed to leverage more features in the network. As shown in Fig. 4 (c), MDF mainly composed of two parts: a multi dimensions feature extractor and a convolutional block attention module (CBAM) [16]. The multi dimensions feature extractor consists two branches: 3D convolution branch and modified anisotropic convolutional branch. Regular 3D convolution branch contains two successive convolution layers with kernel of 3×3×3, and is used for capturing spatial information. The modified anisotropic convolutional branch only uses the 3×3×1 kernel to learn the shape characteristics in the x/y plane. In fact, the modified anisotropic convolutional operations can leverage intra-slice information without increasing the amounts of parameters too much in a 3D network. Finally, the output of two branches is concatenated, which can be formulated as:

$$m_c = Concat\left(Conv\left(3 \times 3 \times k_i\right)(x)\right) \quad i = 1, 3 \tag{2}$$

The CBAM is a lightweight attention module that combine the spatial attention and channel attention, and it can be embedded in other networks easily.

CBAM contains two separate submodules: Channel Attention Module (CAM) and Spatial Attention Module (SAM), and the two submodules work respectively. The CBAM incorporated after the multi dimensions feature extractor is used to focus on the important feature in different channels from the concatenated feature maps. And the SAM is also applied to learn the region of interest which contain potential lesion characteristics. Moreover, the residual connection also applied in the MDF to leverage more feature. Finally, the output of MDF operation is formulated as:

$$Output = \mathbf{F}_{sam}\left(\mathbf{F}_{cam}\left(\mathrm{m}_c\right)\right) + x \tag{3}$$

**Deep Supervision Mechanism** Increasing the depth of the neural network can improve the representation ability, while cause vanishing gradient problem to make the network become difficult to train. Deep supervision is proposed to mitigate such problems. Unlike the previous networks, deep supervision provides integrated direct supervision to the hidden layer instead of only providing supervision in the output layer and passing it back to the earlier layer. This can effectively solve the problems of gradient disappearance and slow convergence. As shown in Fig.3, we introduced deep supervision in the last three decoder blocks of the decoder part. The output layer and the hidden layer can be supervised via deep supervision at the same time, which can jointly improve the gradient propagation to minimize the loss.

### 2.3 Loss Function

In order to alleviate the imbalance problem from multi-class segmentation task, the combination of dice loss and weight cross entropy (WCE) loss is utilized in our network. The loss function can be formulated by the following:

$$\mathcal{L}_{mass} = \alpha\mathcal{L}_{dice} + (1-\alpha)\mathcal{L}L_{wce} \tag{4}$$

where $\alpha$ is the weight and be set to 0.5 in our experiment. And the formulation of $L_{dice}$ and $L_{wce}$ can be describe detail as follow:

$$\mathcal{L}_{dice} = 1 - \frac{2 * \sum_{i=1}^{3}\sum_{n=1}^{N}\left(r_{i_n} * t_{i_n}\right)}{3 * \sum_{i=1}^{3}\sum_{n=1}^{N}\left(r_{i_n} + t_{i_n}\right) + \theta} \tag{5}$$

$$\mathcal{L}_{wce} = \sum_{i=0}^{3} \mathrm{w}_i \sum_{n=1}^{N}\left(r_{i_n}\log\left(t_{i_n}\right) + \left((1 - r_{i_n})\log\left(1 - t_{i_n}\right)\right)\right) \tag{6}$$

Where N denote the voxel number, and i denote the index of each voxel. $r_{i_n}$ and $t_{i_n}$ is the predicted result and the target label of voxel n on category i. And $w_i$ is the weight in the weight cross entropy (WCE) loss.

As shown in Fig. 4, the deep supervision is adopted for the multi outputs in the networks. The outputs from various scales are up-sampled to the original image size. Hence, the final loss in out training stage is formulated as:

$$\mathcal{L}_{total} = \frac{1}{3} \sum_{l=2}^{l=4} \mathcal{L}_{mass}\left(R_l, T\right) \qquad (7)$$

## 3   Experiment

### 3.1   Datasets

The abdominal CT of 300 patients from KiTS21 challenge are applied for training and evaluation in our experiment. Kidney Tumor Segmentation (KiTS) dataset was collected from either an M Health Fairview or Cleveland Clinic medical center between 2010 and 2020. The dataset provides the segmentation labels which contain four classes: (i) background, (ii) kidney, (iii) tumor and (iv) cyst. And the ground-truth of each CT are annotated by professional medical experts. It should be noted that excluding and modifying training cases was explicitly permitted. Therefore, we excluded 10 cases, which some slices are contaminated and loss some information so that unable to load and train by our network. The IDs of these cases are 52,60,65,66,91,111,115,135,140,150. Finally, the rest 290 CT scans were randomly split into training set and validation set with a radio of 4:1. Due to use cross-validation would force the computation time to be multiplied by the number of folds, we only keep a validation set rather than using cross-validation. In order to strike a balance between training enough data and being able to predict our errors, 20% is selected as the account of validation. In the experiments, we train the model and fine-tune hyper-parameter on the training set, and the validation set was used for chose the model with best results.

### 3.2   Pre-processing and Post-processing

One challenge in kidney and tumor segmentation is the unclear boundary. The mitigate the problem, Gaussian filter is applied, then CT voxel intensity of the filtered images is clipped into [-100,400]. Finally, the images are normalized via the z-score normalization. Considering the limit of GPU memory, a sliding window technique is adopted to crop the whole CT image into smaller patches. In the first stage, the size of patches is 256×256×16 and the stride is [128,128,8]. For the second stage, we cropped patches in the VOI after mask inference. And the size of final training patches is 96×96×64 with a stride of [48,48,32].

In the post-processing, the connected component analysis is utilized to remove the unconfident candidates and keep the largest two components as left and right kidney.

### 3.3   Training and Implementation details

The proposed networks were implemented using Python based on the Pytorch and experiments were performed on a computer with a single GPU (i.e., NVIDIA GTX 1080 Ti) and Linux Ubuntu 18.04 LTS 64-bit operating system. We use

Adam optimizer ($\beta1 = 0.9$, $\beta2 = 0.999$) with initial learning rate 1.0-04 for optimization in the training stage. And the batch-size and training epochs are set as 2 and 50 respectively.

### 3.4   Metrics

Our method was evaluated by its Sørensen-Dice score and surface Dice. The Sørensen-Dice are defined by the following formulas:

$$D(P, G) = \frac{2 \times |P \cap G|}{|P| + |G|} \tag{8}$$

where P and G represent the predicted segmentation results and the ground truth, respectively. It should be noted that the quantitative results in our experiment are calculated by the evaluation codes, which provided by the challenge organizations.

## 4   Results and Discussion

Table 1 shows the results of the ablation study for our proposed method and highlights the effect of each component applied to the model on the segmentation results. We evaluate the performance of each component by removing the multi-dimension feature module (MDF) and deep supervision (DS), respectively. We exploit Sørensen-Dice and surface Dice as the evaluation metric and finally report the average of all cases.

From the table it is observed that, during validation, the proposed method achieves the Sørensen-Dice score of 0.9304, 0.5729 and 0.563 for kidney, masses and kidney tumor respectively by involving multi-dimension feature module (MDF) and deep supervision mechanism. Similarly, our network without MDF got separately Sørensen-Dice score of 0.9321, 0.5535 and 0.5463 for the three-class regions. It can also be found that, without incorporating DS in the proposed architecture, Sørensen-Dice score is reduced to 0.9306, 0.5519 and 0.5262 respectively for kidney, masses and kidney tumor.

The qualitative results of KiTS21 dataset on our proposed model is shown in Fig.5. The first column shows the ground truth of input images. The rest columns show the results of our ablation study. In the output images, the green and blue colored spot are tumor region and cyst region, whereas the red color spot is the kidney region. From the qualitative results it is observed that the efficacy of our proposed network.

## 5   Conclusion

In this paper, we described a two-stage cascade framework to obtain kidney, kidney tumor and cyst, respectively. The complex multi-class segmentation task is translated into two subtasks. We designed two networks to implement the subtasks and named them Kidney-Net and Masses-Net, respectively. Kidney-Net is

**Table 1.** Ablation study on the KiTS validation dataset for the multi-dimension feature module (MDF) and deep supervision (DS) into the baseline framework.

| Method | Dice kidney | Dice masses | Dice tumor | SD kidney | SD masses | SD tumor |
|---|---|---|---|---|---|---|
| Ours | 0.9304 | 0.5729 | 0.563 | 0.8722 | 0.4006 | 0.3987 |
| Ours w/o MDF | 0.9321 | 0.5535 | 0.5463 | 0.8758 | 0.3765 | 0.3784 |
| Ours w/o DS | 0.9306 | 0.5519 | 0.5262 | 0.8723 | 0.3783 | 0.3687 |
| U-Net | 0.9307 | 0.5333 | 0.5209 | 0.8722 | 0.3658 | 0.364 |

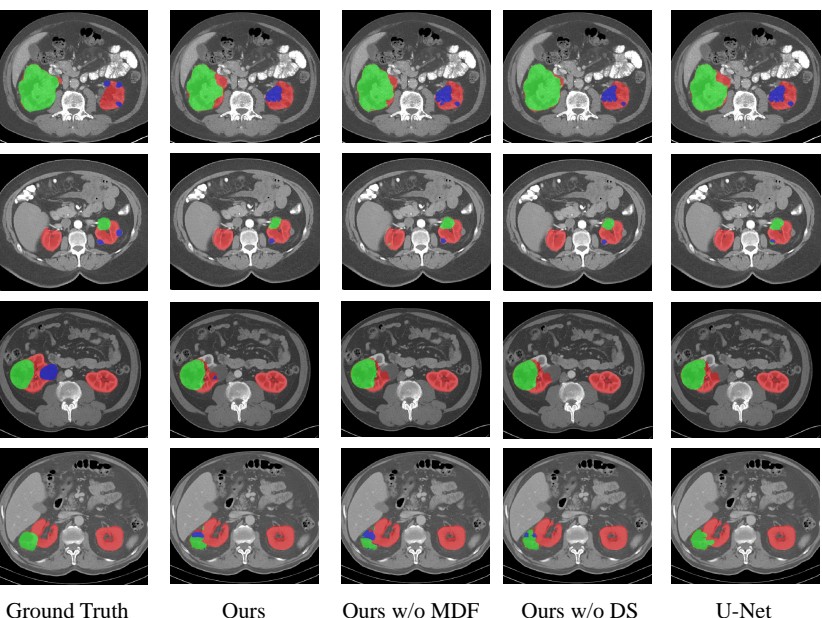

Ground Truth      Ours      Ours w/o MDF      Ours w/o DS      U-Net

**Fig. 5.** Examples of segmentation results. Each row denotes one patient, and from left to right, each column represents the ground truth and different predictions by our method, our network without MDF, our network without DS and U-Net, respectively. The red mask and green mask represent separately kidney area and masses (tumor and cyst) area, while the blue mask represents cyst area.

used for kidney segmentation while Masses-Net is applied for masses (tumor and cyst) segmentation. We merge the outputs from two networks and the merge result is refined via a post-processing method. In order to leverage more space and context information, a multi-dimension feature (MDF) module is embedded into the Mass-Net. And convolutional block attention module (CBAM) also applied to learn important feature. In addition, deep supervision mechanism is utilized for improving segmentation accuracy.

Our method segment three critical organs on KiTS21 validation dataset with the Sørensen-Dice of 0.9304, 0.5729 and 0.563, respectively. While our method

has shown effectiveness on the validation set, we will continue to work on further optimization of the network in the future.

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
