# OpenReview forum: "Kidney and kidney tumor segmentation using a two-stage cascade framework"
_MICCAI.org/2021/Challenge/KiTS — Submitted to KiTS21 Challenge_

### Official Review · Reviewer_FgN2 · 2021-08-30

**Rating:** 7

**Review:**

The authors present a multi-stage approach to the problem that starts by segmenting the kidneys and then segments the tumors from within them. The approach makes use of attention mechanisms to try to incorporate context information in the fine-grained segmentation of the small VOIs. The paper is long-winded and includes several figures and tables that are employed effectively to support the authors' arguments.

The authors mention that they excluded ten cases from their training/validation but don't provide much information as to why. They should expand on this and explain what they mean when they say "unable to load and train by our network". Were they too large? Why couldn't they have been split and done as patches.

The authors also don't say which method they used to aggregate the individual instance segmentations into composite segmentations that can be used for training. Did they use majority voting? If so, they should metion this explicitly.

---

### Official Review · Reviewer_kVRk · 2021-08-30

**Rating:** 8

**Review:**

### Overall

- This is paper does an excellent job providing an in-depth description of this team's approach and preliminary results. I have no comments other than to say that the authors should be sure to include their final test set results once they are known.

### Introduction

- Looks good

### Methods

- Looks good

### Results

- Looks good

### Discussion and Conclusion

- Looks good

---

### Decision · Program_Chairs · 2021-08-30

**Decision:**

Minor Revisions

**Comment:**

Please address the reviewer comments and resubmit